# Targeting Heat Shock Proteins in Malignant Brain Tumors: From Basic Research to Clinical Trials

**DOI:** 10.3390/cancers14215435

**Published:** 2022-11-04

**Authors:** Aisha Babi, Karashash Menlibayeva, Torekhan Bex, Aidos Doskaliev, Serik Akshulakov, Maxim Shevtsov

**Affiliations:** 1National Centre for Neurosurgery, Turan Ave., 34/1, Astana 010000, Kazakhstan; 2Personalized Medicine Centre, Almazov National Medical Research Centre, 2 Akkuratova Str., 197341 Saint Petersburg, Russia; 3Laboratory of Biomedical Nanotechnologies, Institute of Cytology of the Russian Academy of Sciences, 194064 Saint Petersburg, Russia; 4Department of Radiation Oncology, Klinikum Rechts der Isar, Technical University of Munich, 81675 Munich, Germany

**Keywords:** heat shock proteins, small HSPs, Hsp27, Hsp40, Hsp70, Hsp90, inhibitors, glioblastoma, brain tumors, prognostic marker

## Abstract

**Simple Summary:**

Heat shock proteins (HSPs) play an important role in cellular metabolism and therefore are highly expressed in malignant brain tumors. In the current review, the authors assessed the prognostic value of HSPs in neuro-oncology and the possibility of employing these proteins as a target to develop novel therapeutic approaches. Indeed, several preclinical studies indicate the therapeutic potential of small molecular inhibitors of HSPs for targeting brain tumors when being applied as a monotherapy or in combination with other treatment approaches.

**Abstract:**

Heat shock proteins (HSPs) are conservative and ubiquitous proteins that are expressed both in prokaryotic and eukaryotic organisms and play an important role in cellular homeostasis, including the regulation of proteostasis, apoptosis, autophagy, maintenance of signal pathways, protection from various stresses (e.g., hypoxia, ionizing radiation, etc.). Therefore, HSPs are highly expressed in tumor cells, including malignant brain tumors, where they also associate with cancer cell invasion, metastasis, and resistance to radiochemotherapy. In the current review, we aimed to assess the diagnostic and prognostic values of HSPs expression in CNS malignancies as well as the novel treatment approaches to modulate the chaperone levels through the application of inhibitors (as monotherapy or in combination with other treatment modalities). Indeed, for several proteins (i.e., HSP10, HSPB1, DNAJC10, HSPA7, HSP90), a direct correlation between the protein level expression and poor overall survival prognosis for patients was demonstrated that provides a possibility to employ them as prognostic markers in neuro-oncology. Although small molecular inhibitors for HSPs, particularly for HSP27, HSP70, and HSP90 families, were studied in various solid and hematological malignancies demonstrating therapeutic potential, still their potential was not yet fully explored in CNS tumors. Some newly synthesized agents (e.g., HSP40/DNAJ inhibitors) have not yet been evaluated in GBM. Nevertheless, reported preclinical studies provide evidence and rationale for the application of HSPs inhibitors for targeting brain tumors.

## 1. Introduction

Malignant brain tumors, particularly multiforme glioblastoma (GBM), are a challenging diagnosis due to their deep location in the brain, aggressive behavior as well as dismal prognosis with a high mortality rate [1] and low quality of life [2] among the patients (recent CNS tumors classification reviewed in [3]). Generally, the treatment for malignant brain tumors includes surgical resection followed by radiotherapy, chemotherapy with temozolomide, and palliative care. However, the success rate of this treatment scheme remains low; the survival rate constitutes 14.6 months [4]. The addition of the Tumor-Treating Fields (TTF) to standard therapy, which could alter the GBM cell division, resulted in improved progression-free survival and overall survival (of 20.9 months) [5]. One of the promising treatment strategies could be based on the application of targeted therapies [6,7]. Apart from widely used key targets in dysregulated signaling pathways in GBM (e.g., TP53, tyrosine kinase receptors, PI3K/AKT/mTOR pathway, etc.), heat shock proteins (HSPs) represent a promising target for developing novel therapeutic approaches [8,9].

Heat shock proteins constitute a family of conserved proteins in both prokaryotic and eukaryotic organisms that play an important role in the regulation of polypeptides and protein folding/unfolding (i.e., proteostasis), regulation of apoptosis and autophagy, and protection of cells from various stresses (including hypoxia, heat shock, ionizing radiation, etc.) (Table 1) [10,11,12]. Therefore, the expression of HSPs is significantly increased in tumor cells as compared to normal cells [13]. HSPs are classified according to their molecular weights constituting various families—HSPH (HSP110), HSPC (HSP90), HSPA (HSP70), DNAJ (HSP40), small HSP (HSPB), human chaperonin families CCT (TRiC), and HSPD/E (Table 1) [14]. HSPs play a significant role in tumor cell proteostasis and evasion of apoptosis; they are involved in cancer cell division, DNA repair mechanisms, invasion, and metastasis [10,12,15,16].

The transcription of genes of heat shock proteins is activated by heat shock factors (HSF), represented by dimeric structures recognizing the sequences -AGAAN- (Figure 1) [17]. Under normal conditions, HSF monomers are partially inactivated by binding to HSPs in the cytosol. Under stress, the affinity of HSP to denaturing proteins is higher, so HSF1 dissociate from the complex and are transported to the nucleus, where they induce transcription of HSP genes (including Hsp27, Hsp40, Hsp70, and Hsp90). Thus, the activity of synthesis of new chaperones depends on the state of the cell and external influences.

Major chaperones, HSP70 and HSP90, function in the cytosol as refolding proteins, while small HSPs, including Hsp27, prevent aggregation of unfolded polypeptides and proteins [18,19]. A family of HSP40 containing a “J-domain” consists of three subclasses depending on the level of conservation of their J-domain [20]. The main role of HSP40 is the regulation of HSP70 activity [21]. Additionally, Hsp40 facilitates the binding of Hsp70 to the Hsp90-Hop (heat shock organizing protein) complex for further control of the protein folding [22,23,24]. Taking into consideration the protective role of HSPs, the overexpression of chaperones was reported in various types of solid and hematological malignancies [25,26,27,28,29]. High expression of HSPs also correlated to the chemo- and radioresistance of tumors indicated the possibility of employing these proteins as prognostic and/or diagnostic markers [26,30,31,32,33,34].

Many studies have reported that HSPs play an important role in regulating apoptosis and participating in autophagy [35,36,37,38]. Thus, Hsp27, a member of the small chaperones’ family, inhibits apoptosis by controlling the extrinsic apoptotic pathway [35]. It has also been reported that Hsp27 can directly associate with cytochrome c in the cytosol and inhibit the apoptosome formation [36]. Members of HSP70 and HSP90 families have also been reported to be involved in a pathway of apoptosis. Hsp70 can bind the death receptors and inhibit the TRAIL-induced assembly of the DISC at the pre-mitochondrial level [37]. Hsp90 has been reported to block apoptosis at a post-mitochondrial level. The authors demonstrated that HSP90 hampered the caspase activation [38].

Other effects of HSPs that could be employed for the development of anti-cancer therapies is their potential antiangiogenic properties which were reported by several studies [39,40,41,42]. HSP90 inhibitors exert antiangiogenic effects by affecting the PI-3K/AKT and eNOS signal transduction pathways in endothelial cells [43]. It was shown that the connection of HSPs to the eNOS pathway and the activation of PI-3K/AKT pathways encourages Ang-1-induced phosphorylation of eNOS and consequent angiogenesis [44].

Apart from the involvement of HSPs in proteomic homeostasis, it was demonstrated that these proteins exert immunomodulatory activities (reviewed in [45,46]). Thus Grp94(Gp96)-chaperoned peptides were shown to induce a protective T lymphocyte-mediated immune response that involved the uptake of HSP-peptide complexes by antigen-presenting cells with subsequent cross-presentation of these peptides on MHC I class molecules to cytotoxic CD8+ T-cells [45,47,48]. These immune properties of the HSPs were further employed for the generation of anti-cancer vaccines [49,50].

In the current review, we aimed to assess the prognostic/diagnostic values of HSPs expression in malignant brain tumors. Additionally, we analyzed the future perspectives of therapeutic approaches designed to target the HSPs in brain tumors, particularly when the combined treatment strategies were proposed.

## 2. Expression of HSPs in Brain Tumors

### 2.1. Small HSPs

Among small heat shock proteins, Hsp27 is one of the most studied. Immunohistochemical analysis performed by several research groups has revealed a difference in the expression of HSP27 in different grades of glial tumors. The strongest expression was found in glioblastomas, followed by anaplastic astrocytomas, and only moderate expression in astrocytomas [51,52,53,54]. However, the extent of Hsp27 expression in astrocytomas is not definitive, as an immunohistochemical analysis found adjacent para-cancerous brain tissue to have higher expression of the chaperone than low-grade (grade II) glioma tumors [55]. No differentiation in the expression of the chaperone was found between recurrent and primary astrocytic tumors, nor did it vary by the sex of the patients [51]. The factor of age in the chaperone expression, however, was not as clear, as Assimakopoulou et al. did not find an association [51], while in the study by Cai et al., elder patients had higher Hsp27 and phosphorylated p-Hsp27 expression [53]. They also reported no correlation of Hsp27 expression with the survival of the patients [53]. Intriguingly, subsequent subgroup analysis of patients (n = 421) revealed that expression of phosphorylated p-Hsp27 (but not Hsp27) strongly correlated with ATRX (ATP-dependent helicase ATRX, X-linked helicase II) loss (ATRX^-^) and the isocitrate-dehydrogenase IDH1^R132H^ mutation (Figure 2). This subgroup of patients showed an intermediate overall survival (15.0 vs. 13.1 months, *p* = 0.045). Better sensitivity for standard therapy was also reported in p-Hsp27^+^ GBM patients without the IDH1 mutation and ATRX loss (26.3 vs. 15.5 months, *p* = 0.008). As was shown previously, under different conditions, including stress, Hsp27 could be phosphorylated, which leads to its heterogenous oligomerization with subsequent loss of chaperone activity [56,57]. Presumably, this post-translational modification of p-Hsp27 could influence the tumor cell sensitivity towards applied therapies.

Another study employing quantitative proteomic analysis found Hsp27 (HSPB1) expression to be significantly higher in short-term survival patients when compared to long-term survival patients (Figure 3) [54].

In a study that examined Hsp27 expression along with Activator protein 1 (AP-1) transcription factor expression, it was found that HSP27 was only expressed in tumors that had c-Jun and c-Fos co-expression, which are major parts of AP-1. The authors conclude that there might be a relation between HSP27 and AP-1 activation, noting the correlation between AP-1/HSP27 co-expression and the rising grade of the tumor malignancy [58]. This suggests the involvement of Hsp27 in tumor cell survival, as AP-1 plays a major part in cell proliferation and growth [59]. Hsp27 also has an established client relationship with Androgen Receptor (AR) that is excessively expressed in the glioblastoma [60], which might explain the sex disparities in glioblastoma patients. By targeting HSP27, Li et al. demonstrated the potential for AR degradation by employing the HSP27 inhibition [61].

Given the involvement of Hsp27 in the processes ensuring tumor cells survival and with evidence of an increase of Hsp27 expression with the increase of glial tumor grade, the chaperone could serve as an indicator of tumor progression. However, more studies are needed to determine if Hsp27 could be used for the prognosis of the survival outcomes in the patients.

α-B-crystallin expression in glial tumors is more ambiguous, with some studies identifying high expression in the glioblastoma [62,63] and others finding heterogeneity in expression of the chaperone varying from high to undetectable [64,65,66]. Furthermore, a study found the expression of α-B-crystallin to be higher in grade I and II astrocytomas than in grades III and IV, although highly migratory glioma cells overexpressed α-Bc and were resistant to apoptosis [67]. However, no association between the patients’ survival was found with α-B-crystallin expression [66].

Immunohistochemical analyses showed higher expression of Hsp10 in astrocytoma tissue. Moreover, through multivariate analysis, Hsp10 was established as an independent factor negatively associated with poor prognosis when tumor grade, treatment received, tumor size, age, sex, and poly (ADP-ribose) polymerase (c-PARP) proteins were accounted for. The authors came to the conclusion that high expression of Hsp10 leads to inhibition of apoptosis in tumor cells and consequential poor survival of the patients (Figure 4) [68].

Small heat shock protein expression in brain tumors other than glial has been studied less frequently. Among the studied tumor types, medulloblastomas showed only a weak reaction to Hsp27 in immunohistochemical studies [63,69,70], and no association was established with the survival of the patients [70].

Anaplastic meningiomas were more reactive, with 24% of undifferentiated by-grade samples stained for Hsp27 antibodies [69]. Another study of meningiomas found significantly more intense staining of anaplastic meningiomas with HSP27 when compared to their lower-grade counterparts. However, no such difference was observed with α-B-crystallin, where both benign and malignant meningioma types showed weak reactivity [63].

Similarly, anaplastic ependymomas had intense staining for HSP27 and occasional reaction with α-B-crystallin [63], which is in line with the findings by Kato et al., who detected a minor expression of α-B-crystallin in ependymomas [69].

On the contrary, gangliogliomas did not express HSP27 and α-B-crystallin at the glial component of the tumor. Embryonal tumors (i.e., medulloblastoma and ependymoblastoma) showed weak reactions to both proteins; however, the sample size in the study was small (5 and 1, respectively) [63].

### 2.2. HSP40

The family of 40 kDa heat shock proteins and their role in brain tumors is still comparatively under-investigated. A recent study by Liu et al. has found an upregulation of DNAJC10 mRNA, a member of the Hsp40 protein family, in both low-grade gliomas (WHO grade I and II) and high-grade gliomas (glioblastomas) in comparison to normal brain tissue [71]. The DNAJC10 protein expression increased with the increase of WHO glioma tumor grade. An association was found with DNAJC10 increased expression and an MGMT unmethylated status, IDH-wild type, and 1p/19q non-codeletion. Moreover, protein overexpression was associated with a poor survival prognosis for both LGG and glioblastomas (Figure 5) [71].

This finding is supported by the study of Sun et al., that found upregulation of DNAJC10 was associated with shorter survival time in patients [72]. The same study has also found overexpression of DNAJB6 and DNAJB1 to lead to worse survival outcomes and overexpression of DNAJA4, DNAJC6, and DNAJC12 to lead to better survival outcomes [72].

Expression of Hsp40 was found in medulloblastomas, but no difference in expression was observed between different medulloblastoma types [73]. Additionally, epigenetic silencing of methylation-controlled J protein *MCJ (DNAJD1)* through extensive methylation of CpG island was found, which indicates that the inactivation of MCJ might be involved in the tumor formation [74].

In conclusion, Hsp40 proteins as predictive and prognostic factors for glial tumors are a promising area. However, more research is needed to identify its prognostic and diagnostic values.

### 2.3. HSP60

Hsp60 is also abundant in brain tumors and is expressed higher than in normal tissues [75,76]. However, immunohistochemical analysis showed that the expression levels do not differ between glial tumor grades [77]. Neither did the Hsp60 expression in brain tumors differ from other disease tissues, such as cerebral infarct, amyotrophic lateral sclerosis, multiple sclerosis, and acute disseminated encephalomyelitis [78]. A more recent study by Hallal et al. that used cavitron ultrasonic surgical aspirator (CUSA) liquid tumor biopsies containing exosomes for analysis of the chaperonins showed differing results [79].

The Hsp60 family (also called chaperonins) form double-ring oligomeric protein complexes consisting of 60 kDa subunits, with a central cavity that facilitates adenosine triphosphate–dependent folding of polypeptides. The cytoplasmic chaperonin CCT group includes the eukaryotic cytoplasmic chaperonin CCT (chaperonin containing the T-complex polypeptide–1 [TCP1]) families [80]. Proteins belonging to T-ComplexProtein 1 Ring Complex (TRiC), a member of the Hsp60 protein family, were found to be more abundant in glioblastomas compared to low-grade gliomas. Specifically, TCP1, chaperonin containing tailless complexes—CCT2, CCT5, CCT6A, and CCT7-- had a statistically significant difference in expression. This was further checked with TCGA data in silico. All TRiC subunits were significantly increased compared to normal brain tissue, and CCT2, CCT3, CCT5, CCT6A, and CCT8 were increased relative to grade II–III astrocytomas as well. CCT6A specifically had the highest effect on the change of gene expression in glioblastoma samples, and it had an inverse relation to the patient survival [79]. Although elevated expression of the Hsp60 protein family is not unique to glial tumors, there is a potential for using it as a biomarker for tumor malignancy. However, more studies of the family proteins are required to find the optimal candidate.

Through the few studies done on other brain tumors, it was found that meningiomas have elevated levels of Hsp60 [75,77]. Ependymomas did not have a positive immunohistochemical reaction with Hsp60 [75]. Meanwhile, medulloblastomas had increased expression of Hsp60, although it did not associate with the survival outcomes for the patients [73].

### 2.4. HSP70

The expression of Hsp70 family proteins is well-studied in glioblastomas. However, the expression of this chaperone and the effect of protein on tumor survival are ambiguous. Although many studies found the expression of Hsp70 to be elevated in the tumor tissue when compared to normal brain cells [81,82,83], some did not find such differentiation [72,84]. Several studies were performed to understand the difference between recurrent and primary GMB with varying results. Muth et al. found the expression of the extracellular Hsp70 to be significantly lower in the primary GBM compared to the recurrent [81]. No such difference existed with intracellular Hsp70. They also found an elevation of Hsp70 expression to be associated with better prognostic outcomes. These findings contradict those of Thorsteinsdottir et al., who also found elevated expression of Hsp70 (cytosolic, plasma membrane-bound, and extracellular) in tumor tissue compared to healthy; however, expression was much higher in primary glioblastoma than in recurrent GBM, astrocytoma (GII and GIII), and control (epilepsy patients) (Figure 6) [82]. The study also did not find any association with progression-free survival or overall survival or O-6-methylguanine-DNA methyltransferase (MGMT) promoter status. Interestingly, immunoblot analysis in vitro and immunohistochemistry analysis in vivo showed worse survival in patients with high Hsp73 (i.e., constitutive cytosolic Hsc70) expression [66] and higher mRNA expression of *HSPA6*, which was downregulated in gliomas, also resulted in shorter survival [72].

Another study has found a significant increase in cytosolic Hsp70 expression in primary GBM compared to normal tissue and an association of high Hsp70 with better PFS and overall survival [85]. However, the effect on survival was negated in multivariate analysis when MGMT promoter methylation status was accounted for. The confounding effect suggests a relationship between MGMT methylation status and Hsp70 expression, potentially making Hsp70 a biomarker for MGMT methylation. However, the study has its limitations, such as an arbitrary cut-off point of staining 10% or more as an indication of high chaperone expression.

Analysis of HSPA7 (Hsp70) RNA expression showed significantly elevated levels in glioblastoma cell lines (U87MG, U251MG, A172, LN229) compared to normal astrocytes [86]. IDH-mutation types and CpG island methylator phenotype (G-CIMP) tumors had lower levels of HSPA7 expression, and HSPA7 expression was associated with worse survival prognosis after adjusting for other variables such as age, sex, IDH status, MGMT promoter methylation status, G-CIMP status (Figure 7) [86].

Another important protein of the Hsp70 family, GRP78/BiP protein, is significantly overexpressed in glioblastoma, as was confirmed by the immunohistochemistry [87], proteomics [88], western blot analysis of cell lines, and immunohistochemistry of in vivo grown tumors [89], and in vivo in xenografted mice models [90]. It was also found that the expression of GRP78 mRNA is elevated even more in the recurrent GBMs [91]. This effect could be the result of a protective reaction to treatment with temozolomide and radiotherapy that recurrent GBM patients have undergone, as, for example, treatment of glioblastoma cells with cisplatin increased chaperone accumulation by three folds [92].

Expression of GRP78 was not only cytosolic but also on the surface of glioblastoma cells and was highest in GBM (T98G, A172, and U-87 MG) when compared to other tumors, including grade III anaplastic glioma cell line (Hs 683 and U-373 MG). The role of GRP78 in tumor survival was tested with polyclonal N-20 antibodies treatment. Neutralization of the chaperone resulted in the suppression of growth and survival of the tumors. However, the effect was much more noticeable in anaplastic glioma than in the glioblastoma [90].

GRP78 is also predictive of glioblastoma cells’ sensitivity to ubiquitin-like modifier activating enzyme 1 (UBA1) inhibitor TAK-243 [93]. Inhibition of the UBA1 leads to UPR and subsequent cell apoptosis, resulting in longer survival of glioblastoma-bearing mice. Tumors with high levels of GRP78 expression are resistant to treatment with TAK243. The inhibition of UBA1 by the latter causes an increase in GRP78 expression, creating a negative feedback [93]. Overall, GRP78 could be a target for the treatment of high-grade gliomas.

Heat shock cognate protein 70 (Hsc70) is overexpressed in glioma tissues, and its expression increases with the higher grade. Expression was 3-fold higher in grade IV glioma tissues when compared to normal brain tissue. Decreasing the expression of Hcs70 resulted in a slowing of tumor migration and invasion, and its knockdown stimulated apoptosis and reduced cell proliferation [94,95].

Mortalin, another member of the Hsp70 protein family, also has an elevated expression corresponding to increased malignancy in astrocyte tumors and is undetectable in normal brain tissue [96]. The overexpression, however, is not unique to glial tumors but was also detected in other brain tumor types (e.g., meningiomas, neurinomas, pituitary adenomas, and metastases). Thus, an increase in mortalin expression with grade might suggest the involvement of the protein in the malignant transformation [96].

Other brain tumor types were not as thoroughly studied. In medulloblastomas, one study found, among all the studied chaperons, the expression of Hsp70 to be the highest [70]. Another study found a significant increase in Hsp70 expression in large-cell medulloblastoma than in the classic subtype [73]. Meanwhile, immunohistochemical analysis by Rappa et al. did not find an increase in the expression of Hsp70 in the medulloblastomas [77]. None of the studies found an association between the chaperone and patients’ overall survival [70,73].

In meningiomas, expression levels of Hsp70 are elevated [75], regardless of grade, and compared to other tumors, such as neuroepithelial tumors and medulloblastoma, as well as normal brain tissue [77].

### 2.5. HSP90

Hsp90 is also overexpressed in glial tumors [66,84]. Significantly higher levels of Hsp90 expression were found in glioblastomas IDH-wildtype. Moreover, patients with low expression of EGRF and Hsp90, its chaperone, had worse prognostic outcomes [94]. Zhang et al. also found the association of Hsp90 expression with IDHwt status, particularly in low-grade gliomas (LGG) of grades II and III [97]. They also stratified LGG by survival time, taking 36 months as a cutoff point for the better prognosis group. They found elevated expression of *HSP90AB1* to be associated with better survival outcomes. No such analysis was done for WHO grade IV gliomas, defined by them as high-grade gliomas (HGG), which had lower chaperone expression than LGG. They did however find higher expression of *HSP90AA1* and *HSP90AB1* in the recurrent glioblastomas [97].

These results are the opposite of the study that employed a bioinformatics approach querying open repositories and pooling data from several clinical studies with over 1500 samples [98]. The study found mRNA expression of Hsp90 to be the highest in glioblastoma when compared to other tumor types, such as astrocytoma, and oligodendrogliomas, and the chaperon expression has the inverse correlation with disease-free survival (Figure 8) [98]. The same association with survival was found by Sun et al., where HSP90B1 increased expression led to a shorter survival time [72].

Despite the opposing findings on the effect of Hsp90 on the survival of patients, experimental studies show that inhibition of the Hsp90 ATPase domain with Shepherdin, a peptidomimetic antagonist of the complex between the chaperone and survivin, resulted in induction of autophagy in glioblastoma cells and suppression of tumor growth in mice model (immunocompromised CB17 SCID/beige female mice), as well as longer survival [99]. Moreover, inhibition of Hsp90 in glioblastoma cancer stem cell-like cells (T98G) increased their sensitization to the radiotherapy [100]. Clearly, inhibition of Hsp90 has shown a positive effect on survival in vivo and in vitro [101,102]. Acetylation of Hsp90 through HDAC enhanced its chaperone function, promoting the stability of HIF-1alpha, which is involved in cell angiogenesis, cell survival, and tumor invasion. Inhibition of HDAC (with LBH589) resulted in reduced cell proliferation, growth, and microvessel density [103].

Another important protein of the Hsp90 family is TRAP1, a mitochondrial protein found to be associated with chemotherapeutic drug resistance [104], and significantly elevated in glioblastoma cell lines when compared to normal brain tissue [99,105]. Higher-grade glioma tumor tissues have more intense TRAP1 staining than lower-grade tumors and have a higher expression (according to immunoblotting). TRAP1 expression is also associated with a low Karnofsky score and worse overall survival of the patients [106]. Knock-out of TRAP1 in glioblastoma cells sensitized them to temozolomide, decreased migration, induced cell apoptosis, and reduced cell proliferation [105].

Just as with other heat shock proteins, studies on Hsp90 and brain tumors other than glial are rare. It has been found that meningiomas had a positive reaction to Hsp90 with 26 to 45% of the samples reacting on immunohistology tests [75,107]. Medulloblastomas were also positive for Hsp90 to a varying degree [75,107]. However, no association with survival was found in the medulloblastomas [70,73].

## 3. Combination of HSPs with Other Treatment Modalities

Heat shock protein inhibition is emerging as a promising strategy for cancer therapy and demonstrates some promising results [108]. HSPs inhibitors can be effectively used as a monotherapy or in combination with conventional forms of treatment, such as radiotherapy and chemotherapy. A potentially effective treatment outcome has been observed in malignant tumors treated with a combination of HSPs inhibitors with traditional treatment options in malignant tumors [108,109]. Data collected from experimental and clinical studies of HSPs inhibitors and their combination with radio- and chemotherapy, and other treatment modalities in malignant brain tumors are presented in Table 2.

### 3.1. Combination of HSP90 Inhibitors with Other Treatment Modalities

HSP90 inhibitors are among the most studied and offer an effective therapeutic approach to the treatment of cancer. HSP90 family inhibitors have been discussed as an anti-cancer agent in the treatment of malignant brain tumors in combination with other treatment modalities. Several clinical trials have been initiated to test the safety, efficacy, and anti-cancer activity of the combinatorial treatment approaches [116].

Several studies have investigated the 17-allylamino-17-demethoxygeldanamycin (17-AAG or Tanespimycin) inhibitor of HSP90 as a therapeutic agent in combination with standard cancer treatment modalities against glioblastoma [110,112,113]. Thus Sauvageot et al. reported that HSP90 inhibitor 17-AAG might synergize with radiation and have a potential therapeutic effect on the treatment of newly diagnosed glioblastoma [112]. In a study, animals treated with 17-AAG survived longer in comparison with those that did not receive the inhibitor. Intriguingly, 50% of all 17-AAG treated animals survived while all the animals from the control group died [112]. The inhibitor was successful not only in combination with the standard treatment options but also showed the additive radiosensitivity effect in the complex treatment with olaparib, the inhibitor of poly (ADP-ribose) polymerase (PARP). 17-AAG enhances the PARP inhibition effect and thereby increases the radiosensitization of human glioma cells [110].

Another study has investigated the combined inhibitory effect of 17-AGG and ZD1839 (Iressa), an inhibitor of an epidermal growth factor receptor (EGFR) tyrosine, on glioma cells growth [113]. The overexpression of EGFR is a genetic alteration in primary glioblastoma and promotes aggressive tumor cell growth. HSP90 stabilizes Akt and oncogenic forms of mutant EGFR. Akt and mutant EGFR contribute to the growth of glioma cells. HSP90 inhibitors geldanamycin (GA) and Radicicol, also known as the first researched HSP90 inhibitors, decreased the expression of EGFRvIII in glioblastoma cells. The study reported that the effect of Iressa was potentiated in combination with the 17-AAG, which presumably reflects the inhibition of some signaling pathways important to cancer cell growth and survival [113].

Despite the positive outcomes of the 17-AAG combination with other treatment modalities, the therapeutic effect of 17-AAG is limited by its poor permeability through the blood-brain barrier (BBB) and high toxicity. Another inhibitor of HSP90, OSU-03012 (or AR-12), also demonstrated a certain therapeutic potency [117]. Additionally, another Hsp90 inhibitor, 17-DMAG (17-Dimethylaminoethylamino-17-demethoxygeldanamycin) (Alvespimycin), was shown to impact the DNA damage response to radiation and also enhanced tumor cell radiosensitivity [118].

Interestingly, the authors reported that the combined application of OSU-03012 and sildenafil further enhanced the therapeutic effect of inhibitor [119].

One of the inhibitors of the HSP90 family that can penetrate the BBB is HSP990. HSP990, in combination with a PI3-kinase inhibitor, BKM120, and radiotherapy, has demonstrated improved clinical outcomes in patients with glial tumors. HSP990 and BKM120, being effective agents against tumor cells on their own, showed individual induce rates of 75% (BKM120) and 59% (HSP990) for glioma cells death, creating an enhancement in radiosensitivity in combination, which induced 89% apoptosis of cancerous cells, resulting in improved clinical outcomes in patients with primary glioblastoma [114]. Future clinical trials should further prove the therapeutic potency of these agents.

Alongside HSP990, which can penetrate the BBB, another novel inhibitor of the 90-kDa heat-shock protein family—NXD30001-- could also easily cross the BBB and accumulate in the brain. The NXD30001 impairs the DNA Damage Response (DDR) after radiotherapy. In vivo study results show that a combination of radiotherapy and NXD30001 inhibitor extended survival from 31 days to 43 days in glioblastoma mice models (female athymic nude mice implanted with T4302 CD133+ cells) when compared to radiotherapy alone [101]. In line with these results, a previous study has described NXD30001 inhibitor as a molecule that can strike glioblastoma at the core of its drivers of tumorigenesis through apoptosis and degradation of HSP90 client proteins and therefore, can be used in the treatment of glioblastoma [120]. Therefore, the NXD30001 inhibitor could be a potential candidate for the treatment of glioblastoma in combination with radiation therapy.

An in vitro study of another Hsp90 inhibitor, NW457, has demonstrated that a combination of the inhibitor in a low dose with radiotherapy considerably interferes with DNA Damage Response (DDR) in cancerous cells [111]. DDR, in turn, is conducive to sensitization towards radiotherapy, creating a stronger effect of radiotherapy.

Other inhibitors of the HSP90 family that are rarely studied are PU-H71 and NVP-HSP990. The combined effect of PU-H71 and heavy ion radiation has been found to enhance the sensitivity in human cancer cells [121]. While NVP-HSP990 sensitized the U251 glioblastoma cell to an ionizing radiation [122].

### 3.2. Combination of HSP70 Inhibitors with Other Treatment Modalities

The 70 kDa heat shock protein Hsp70 is known to be localized in various cell compartments (including cytosol, nucleus, plasma membrane), and thus several targeting approaches were proposed [25]. Up-to-date several small molecule inhibitors were reported to suppress Hsp70 efficiently. However, the only inhibitor, MKT-077 (1-ethyl-2-[[3-ethyl-5-(3-methylbenzothiazolin-2-yliden)]-4-oxothiazolidin-2-ylidenemethyl] pyridinium chloride) (rhodacyanine dye analog), that entered clinical phase I trial in patients with chemo-resistant solid tumors due to high renal toxicity was halted [123,124]. Other inhibitors (such as flavonoids, 3′-sulfogalactolipids, and 15-deoxyspergulin) demonstrated only a modest anti-tumor therapeutic activity [125,126].

Other Hsp70 inhibitors that showed promising results, particularly when they we employed with other anti-tumor agents, included VER-155008, pifithrin-μ (PES), and MAL3-101 [127,128,129]. Recently, Zhu et al. proposed a nanoplatform (PES-Au@PDA) that could deliver PES, photothermal conversion agent polydopamine (PDA), and radiosensitizer (gold nanospheres, AuNS) for synergistic radiotherapy and photothermal therapy, as well as to visualize the tumor progression employing MRI and/or CT [130]. The preclinical studies demonstrated that PES-Au@PDA particles efficiently induced pro-apoptotic cascades, thus inhibiting tumor progression in human SW1783 glioblastoma-bearing mice (Figure 9) [130].

Another agent, cannabinoid agonist WIN55-212-2, also demonstrated efficacy in vitro (human GAMG and U251 glioblastoma cell lines) and in vivo by altering the expression of p53, cathepsin D and Hsp70 [131]. Another inhibitor, AEAC (N-amino-ethylamino derivate of colchicine), that suppressed substrate-binding and refolding activities of Hsp70 was shown to accumulate in C6 glioma in vivo, further potentiating the effect of doxorubicin that resulted in the reduction of tumor growth rate and prolongation of animal survival by 12 ± 2.2 days [132].

Tumor-specific expression of the plasma membrane-bound Hsp70 provides a possibility to apply, apart from small molecular inhibitors, other targeting agents, including aptamers, peptides, anticalins, and antibodies [25,133,134]. Thus, by the group of Prof. Carmen Garrido (University of Burgundy, Dijon, France), two peptide aptamers—A8 and A17—were designed to bind substrate-binding (SBD) and nucleotide-binding (NBD) domains of Hsp70, respectively. These aptamers specifically inhibited the Hsp70 chaperone activity, thereby promoting the tumor cells’ sensitivity to therapeutic agents that resulted in the regression of subcutaneous tumors in animals [133]. Furthermore, inhibition of Hsp70 by A17 aptamer potentiated the radiosensitizing effects of Hsp90 inhibitor NVP-AUY922 [135]. Application of anti-Hsp70 antibodies conjugated with superparamagnetic iron oxide nanoparticles (SPIONs) also demonstrated high diagnostic potency in the orthotopic C6 model in rats (when MRI was applied) [136]. Furthermore, when nanoparticles were decorated with serine protease granzyme B, which is known to specifically bind membrane-bound Hsp70 [137], magnetic particles could accumulate in the human U87 glioma tissues and induce the apoptotic cell death [138]. The combination of GrB-SPIONs with a single dose (10 Gy) radiotherapy further increased the therapeutic anti-glioma potency of nanoparticles [138].

### 3.3. Combination of HSP40 Inhibitors with Other Treatment Modalities

As was shown previously, HSP47 (encoded by SERPINH1 gene) is overexpressed in primary GBM cells where it functions as a chaperone protein for collagen-promoting tumor cells invasion, angiogenesis, and stem-cell-like properties via the TGF-β pathway [139,140]. Subsequent blockade of this pathway abrogated the tumor-supportive effects of HSP47 [140]. Due to the limited knowledge of the involvement of HSP40 in GBM tumorigenesis, there are few studies demonstrating the therapeutic potency of its inhibition. One of the molecules reported to bind Hsp40 was phenoxy-N-arylcetamides (IC_50_ = 130 nM) that also disrupted the Hsp70-mediated luciferase refolding activity [141]. Other inhibitors were also reported, including benzylidene lactam compound (KNK437, *N*-formyl-3,4-methylenedioxy-benzylidene-γbutyrolactam) [142], natural compound-derived plumbagin derivate (PLIHZ, plumbagin-isoniazid analog) [143], and bioflavonoid quercetin (3,3′,4′,5,7-pentahydroxyflavon) [144]. Presumably, future preclinical trials will elucidate the therapeutic efficacy of HSP40 inhibitors in neuro-oncology.

### 3.4. Combination of HSP27 Inhibitors with Other Treatment Modalities

The member of the small HSPs family is HSP27 (HspB1), and its expression is increased in glioma cells. It has been reported that using resveratrol, a natural compound with antitumor effects, against glioblastoma, in combination with temozolomide might increase the sensitivity of the cancerous cells through the inhibition of HSP27 [115]. A study has demonstrated the potential of Hsp27 siRNA and rosmarinic acid combination applied to human glioma cells as an effective Hsp27 inhibitor and inducer of apoptosis [145].

It was demonstrated that androgen receptor (AR) mutations promote cancer development in glioblastoma, and silencing of AR in glial cells had a potential antitumor effect [60,146]. Thus Li et al. inhibited AR expression in GBM cells by suppressing HSP27 using Compound I that resulted in the suppression of tumor cell growth with IC_50_ values of 5 nM. Subsequent in vivo experiments in the GBM xenograft model demonstrated an acceptable toxicity profile of the compound and therapeutic potency [61]. Furthermore, it should be noted that treatment of GBM cells with temozolomide (TMZ) increases the expression of HSPB1 (HSP27), contributing to the chemoresistance [61,147]. To overcome this challenge, Rajesh et al. applied Friend leukemia integration 1 (Fli-1) as a mediator of HSPB1 and suggested that via the regulation of Fli-1, one could affect the resistance of glioblastoma to TMZ [148]. The authors reported that lumefantrine (FDA-approved anti-malarial medicine which can pass BBB and inhibit Fli-1) has a therapeutic potential against radio- and TMZ-resistant GBM cells [148].

## 4. Conclusions

Molecular chaperones, due to their important role in the physiological processes in cells, are highly expressed in brain tumors, and the level of HSPs expression strongly correlates with a malignancy grade (higher in GBM), invasive potential, as well as resistance to radiochemotherapy. For certain HSPs representatives (i.e., HSP10, HSPB1, DNAJC10, HSPA7, HSP90), a direct correlation between the protein level expression (based on IHC analysis) and poor overall survival prognosis for patients with glial tumors was identified. This indicates the prognostic values of these markers that could be included in the future included into the diagnostic panel in the neuropathological examination of a tumor sample. One of the limitations is an absence of standardized protocols for HSPs detection when mostly IHC analysis (for evaluation of HSPs cytosolic and nuclear expression) and flow cytometry (for detection of HSPs plasma-membrane bound forms) are employed. Moreover, there is still no agreement between the researchers on whether to separately assess the cytosolic and nuclear expression of HSPs on paraffin IHC sections and their prognostic and diagnostic values. Most of the studies report HSPs expression without such a distinction. However, as was previously shown, upon various stress conditions, HPSs do migrate into the cell nucleus (although the list is known about their function in the nucleus) [149,150]. Presumably, in tumor sections, these two patterns of HSPs expression should be evaluated independently.

Taking into consideration the importance of HSPs in the maintenance of proteostasis, signal transduction, regulation of apoptosis, and autophagy inhibition of chaperones might have therapeutic potential. However, most of the reported studies were performed with other malignancies where the inhibitors demonstrated a therapeutic potential either as a monotherapeutic agent or in combination with other treatment modalities. This gives hope for future studies that these inhibitors could also be beneficial when used for targeting brain tumors.

## Figures and Tables

**Figure 1 cancers-14-05435-f001:**
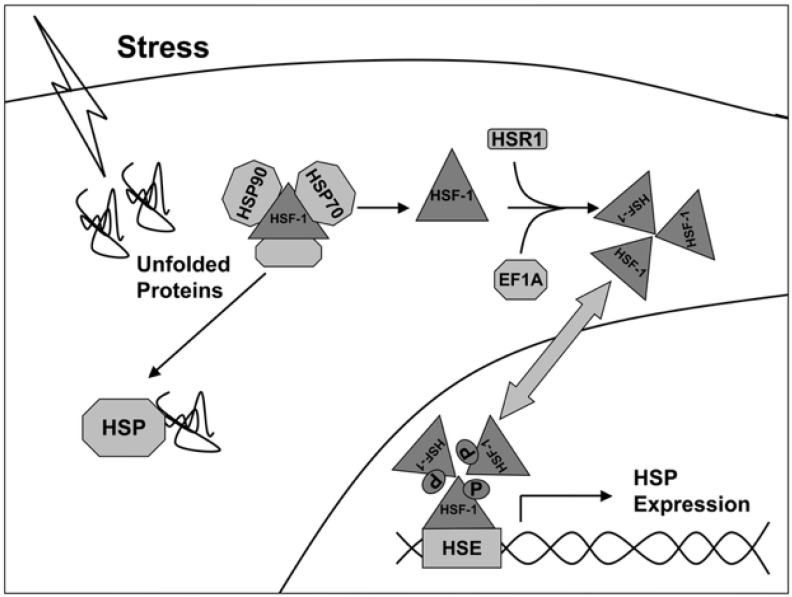
Activation scheme of HSP gene expression under stress conditions [17].

**Figure 2 cancers-14-05435-f002:**
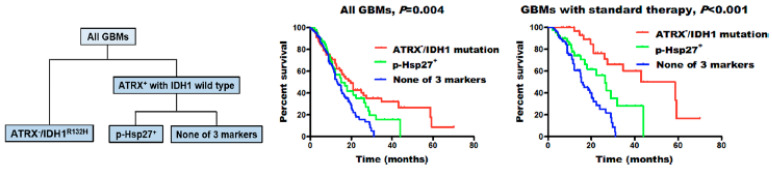
Molecular classification of glioblastomas (GBMs). GBMs were separated into three groups: ATRX− /IDH1R132H, high p-Hsp27 expression (p-Hsp27+), and none of these three markers. Individuals with ATRX− /IDH1R132H showed the longest median survival, those with high p-Hsp27 expression had an intermediate prognosis, and those without any alteration in the three proteins had the poorest survival rates [53].

**Figure 3 cancers-14-05435-f003:**
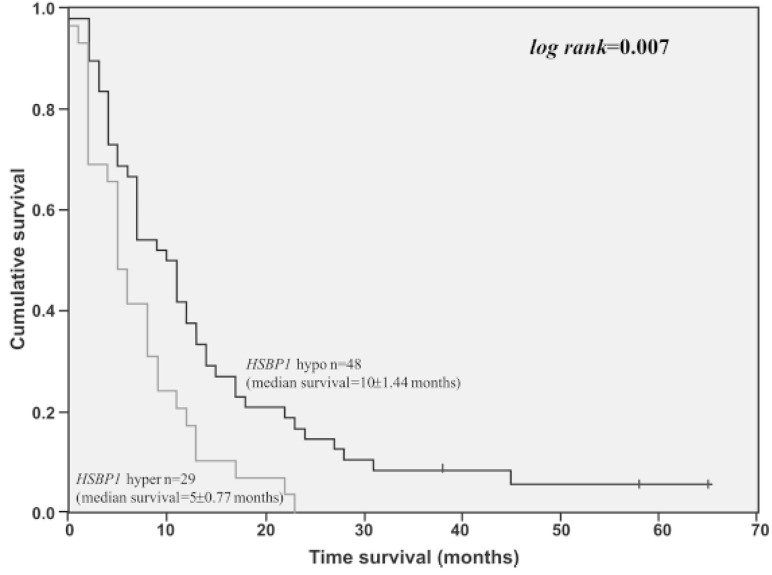
Overall survival time of GBM cases presenting HSPB1 high expression level (3-fold of cut-off value determined by ROC curve) (n = 29) compared to GBM cases presenting lower HSPB1 expression level (n = 48). Log-rank = 0.007 [54].

**Figure 4 cancers-14-05435-f004:**
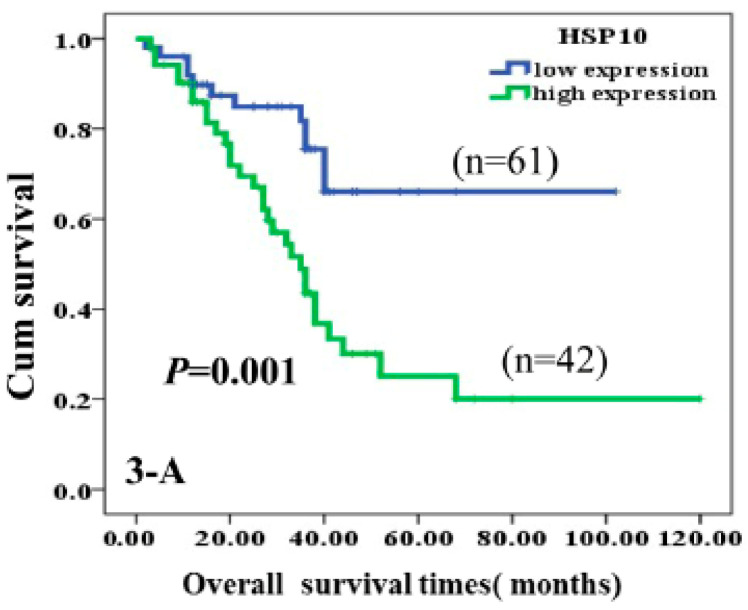
Kaplan-Meier curves according to expression of HSP10 protein divided into high and low expression. High expression of HSP10 was significantly correlated to poor prognosis of astrocytoma patients (*p* = 0.001, two-sided) [68].

**Figure 5 cancers-14-05435-f005:**
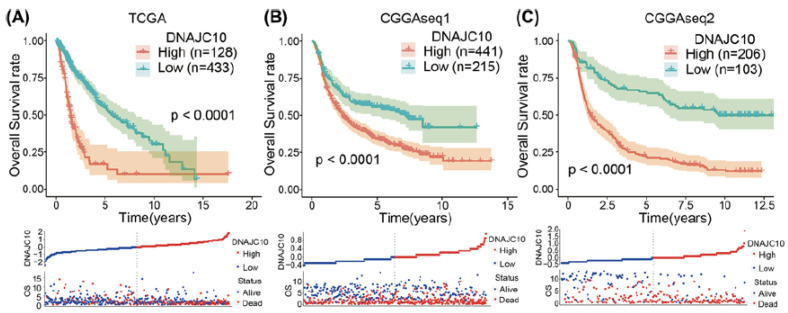
High DNAJC10 expression indicates poor prognosis of glioma patients (**A**–**C**). Kaplan–Meier survival curves indicated that glioma patients with higher DNAJC10 expression levels showed shorter survival time and rate in three independent glioma cohorts (TCGA, CGGAseq1, and CGGAseq2) [71].

**Figure 6 cancers-14-05435-f006:**
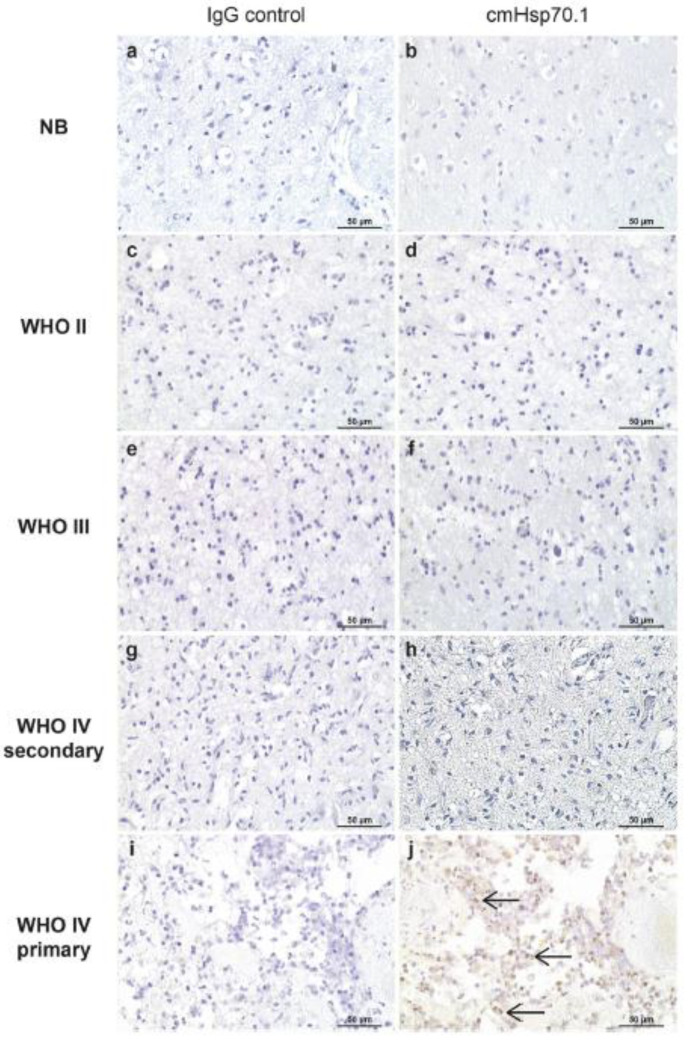
Membrane Hsp70 expression in the non-neoplastic brain (**b**), diffuse astrocytoma (**d**), anaplastic astrocytoma (**f**), secondary GBM (**h**), and primary GBM (**j**) and corresponding negative controls (**a**,**c**,**e**,**g**,**i**) as determined by IHC. Magnification: ×20. Scale bar 50 µm [82].

**Figure 7 cancers-14-05435-f007:**
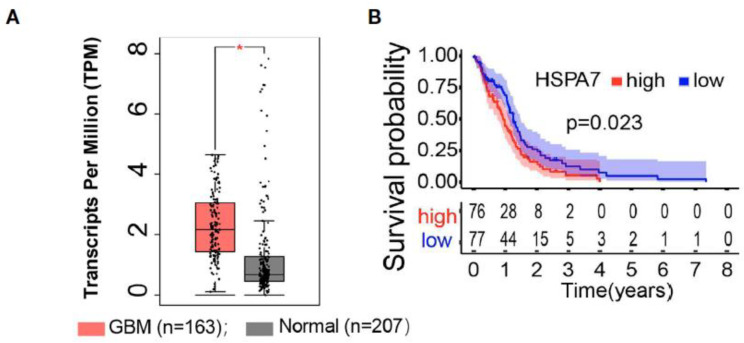
HSPA7 as a novel prognostic factor in GBM. (**A**) The GEPIA database showed that HSAP7 was overexpressed significantly in GBM tissues compared with GETx normal brain tissues. (**B**) Kaplan–Meier survival curves show that HSPA7 is a prognostic risk factor in GBM [86].

**Figure 8 cancers-14-05435-f008:**
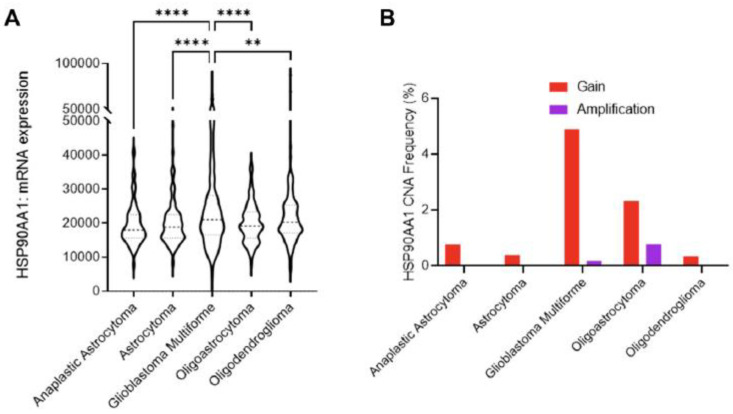
Role of Hsp90 in glioblastoma multiforme (GBM) across CNS tumors. (**A**) HSP90AA1 mRNA expression (RNA Seq V2 RSEM) [log_2_(value + 1)] compared across five clinical studies (cBioPortal). ** *p* < 0.01, **** *p* < 0.0001 analyzed by ordinary one-way ANOVA with Tukey’s multiple-comparison test. (**B**) Copy number alteration (CNA) evaluated across seven clinical studies (cBioPortal) [98].

**Figure 9 cancers-14-05435-f009:**
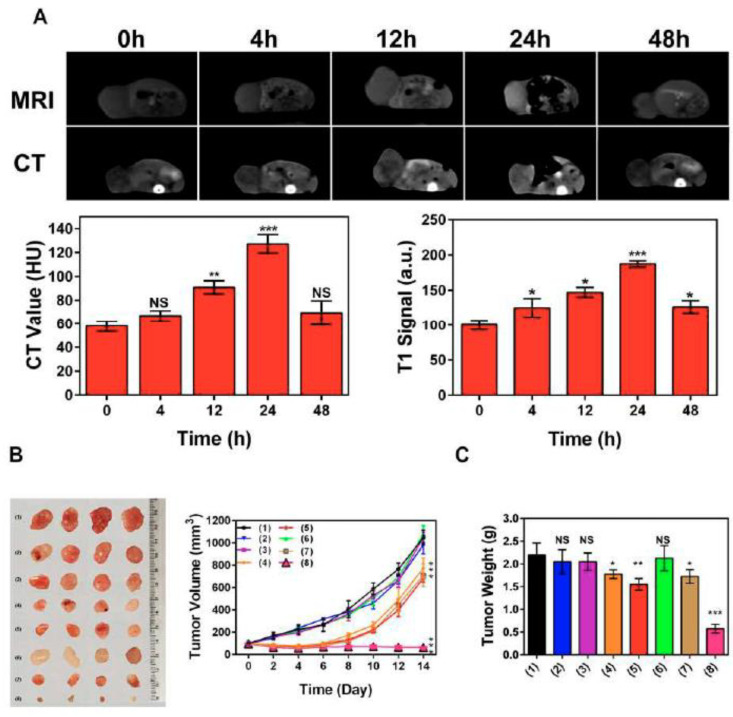
PES-Au@PDA NPs synergistic GBM treatment and multimodal imaging in vivo. (**A**) CT and 3 T1-weighted MR images were acquired at the indicated times (0, 4, 12, 24, 48 h) following intravenous injection of 12 nM•kg^−1^ PES-Au@PDA NPs. (**B**) Ex vivo tumor images and tumor growth curves following different treatments (n = 4 per group): (1) PBS (control), (2) RT (4 Gy), (3) PES-Au@PDA (12 nM•kg^−1^), (4) PES-Au@PDA(12 nM•kg^−1^) + RT (4 Gy), (5) Au@PDA(12 nM•kg^−1^) + RT (4 Gy), (6) laser (808 nm, 1.0 W•cm^−2^, 5 min) + RT (4 Gy), (7) Au@PDA(12 nM•kg^−1^) + laser (808 nm, 1.0 W•cm^−2^, 5 min) + RT (4 Gy), (8) PES-Au@PDA(12 nM•kg^−1^) + laser (808 nm, 1.0 W•cm^−2^, 5 min) + RT (4 Gy). (**C**) Tumor weights on the 14th day 9 after the treatments as indicated in (**B**). HU represents Hounsfield units, RT represents radiotherapy. Experiments 10 were repeated three times, and the data are expressed as the mean ± SEM. * *p* < 0.05, ** *p* < 0.01, *** *p* < 0.001. NS—not significant [130].

**Table 1 cancers-14-05435-t001:** Roles of major HSP family members in cell physiology.

Family	Gene	Cell Localization	Functions
HSP27	*HSPB1*	Cytosol; Membrane-bound; Mitochondria; Cytoskeleton; Nucleus	Transfer of the misfolded proteins to ATP-dependent chaperones and proteasomes; Protection against oxidative stress; Inhibition of apoptosis; Regulation of the cytoskeleton
HSP40/DnaJA	*DNAJA1 (HDJ-2)*	Cytosol; Nuclei; Endosomes; Exosomes; Mitochondria; Ribosomes; ER	Gene expression, translational initiation; Protein folding and unfolding; Translocation and degradation of proteins; Mediating the remodeling of large multiprotein complexes
	*DNAJA3 (Tid1)*		
	*DNAJA4*		
HSP40/DnaJB	*DNAJB1*		
	*DNAJB4 (HLJ1)*		
	*DNAJB6*		
	*DNAJB8*		
	*DNAJB9 (MDG1)*		
HSP40/DnaJC	*DNAJC6*		
	*DNAJC12 (JDP1)*		
	*DNAJC25*		
HSP60	*HSPD1*	Mitochondria; Cytosol; Cell surface; Vesicles	Folding, translocation, assembly of native proteins; Replication and transmission of mitochondrial DNA; Thermotolerance; Intracellular protein trafficking; Peptide-hormone signaling; Resistance to stress-induced apoptosis
HSP70	*HSP72*	Cytosol; Membrane-bound; Nucleus	Folding and transport of newly synthesized polypeptides and aberrant proteins; Assembly of multi-protein complexes; Protection of cells against damage; Immunomodulatory activity (cross-presentation of immunogenic peptides, chaperokines function; stimulation of innate immune responses)
	*HSPA6*		
	*HSC70*		
	*Mortalin*		
	*GRP78*		
HSP90	*HSP90*	Cytosol; Nucleus; Mitochondria; ER; Membrane-bound	Protein synthesis, folding and degradation; Assembly of multiprotein complexes; Integrity of signaling pathways; Functional activation of steroid hormone receptors; Resistance to stress-induced apoptosis; Immunomodulatory activity
HSP110	*HSP110*	Cytosol; Nucleus	Protein folding and disassembly of protein aggregates; Resistance to stress-induced apoptosis;

**Table 2 cancers-14-05435-t002:** A combination of HSPs inhibitors with anti-tumor therapy in malignant brain tumors.

Brain Tumor	HSP	First Therapy/Treatment	Second Therapy	Outcome	Reference
Glioblastoma	HSP90	Inhibitor 17-AAG	Radiation, Chemotherapy	Enhanced the radio sensitizing activity	Dungey, Caldecott and Chalmers, 2009 [110]
Glioblastoma	HSP90	Inhibitor NW457	Radiotherapy	Sensitization towards radiotherapy	Orth et al., 2021 [111]
Glioblastoma	HSP90	Inhibitor NXD30001	Radiation	Increased survival in a glioblastoma model	Chen et al., 2020 [101]
Glioblastoma	HSP90	17-AAG	Radiation	17-AAG can radiosensitize, it has a slight antagonistic effect on growth inhibition with temozolomide	Sauvageot et al., 2009 [112]
Malignant gliomas	HSP90	17AAG	ZD1839 (Iressa)	Impacts cancel cell growth and survival	Premkumar, Arnold and Pollack, 2006 [113]
Glioblastoma	HSP90	HSP990	BKM120, radiation	improve clinical outcome	Wachsberger et al., 2014 [114]
GBM	HSP27		Resveratrol	Enhance the therapeutic effect of resveratrol	Önay Uça and Şengelen, 2019 [115]

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
