# Peer review of "Targeting Heat Shock Proteins in Malignant Brain Tumors: From Basic Research to Clinical Trials"

_cancers, 2022, doi:10.3390/cancers14215435_

Round 1
Reviewer 1 Report
The review paper by Babi et al. reviewed the expression of HSP family proteins in brain cancers, as well as their potential role in disease prognosis and as a drug target for treatment. The paper is well written in English language. However, some contents/organization/details can be revised to improve the quality of the review paper. Please see the comments below, mainly based on the order of the article:
1. Line 54, the term “heat shock proteins” is first mentioned here, so the abbreviation of HSP should be here, not in line 56.
2. Lines 61-63, the classification of HSP needs to be more clearly described with additional information, such as adding family and gene names. Adding a summary table of HSP family should be helpful. Please refer to this review article: https://pubmed.ncbi.nlm.nih.gov/28012700/.
3. Lines 64-66. The sentence is confusing. Please rewrite.
4. Since the expression of HSPs is mentioned in different types of brain tumors (such as glioblastomas, astrocytoma, etc.), it will be good to provide an introduction of brain tumors such as the classification, grade, prognosis, treatment, etc., especially if there is an association with HSPs. A summary table about the relationship between HSP and brain tumors will be helpful. Or the table can be combined with Table 4.
5. “et al.” should be italic.
6. Line 103, p-HSP27 means phosphorylated HSP27, right? The abbreviation of it is mentioned later line 106. It should be introduced at the first place. Why study of the p-HSP27 is important (needs rationale if it is mentioned)? Any particular role of p-HSP27 in brain tumors compared to regular HSP27.
7. Some abbreviations need to give their full name to help readers to understand, such as ATRX, IDH (line 107), PARP (line 149), MCJ (line 195), CCT (line 209), MGMT (line 237) etc. Providing a list of abbreviations will be helpful.
8. Lines 121-123, figure legend is hard to be understood. Do they miss the unit of 23.28?
9. Immunochemical analysis of HSPs is mentioned in multiple places in the manuscript. Where are HSPs expressed in the brain or brain tumors? Providing an image along with the survival curve should be helpful.
10. Line 162, needs empty space before “In conclusion”.
11. Line 200, it says Hsp60 is expressed higher in tumors than normal tissues. Then it says no difference between other disease tissues (lines 202-204). Does this mean HSP60 is not overexpressed in these diseases? What is the purpose of stating this?
12. Line 209, need the introduction of the TCP, CCT members. Are they family members of Hsp60?
13. Line 239, what is the Hsp73? The subfamily of Hsp70, HSC70 is ~73kDa? Or misspell of Hsp70?
14. Line 240, gene name HSPA6 should be italic.
15. Line 250. HSPA7 is a pseudogene. Please see https://pubmed.ncbi.nlm.nih.gov/11072087/, https://www.genecards.org/cgi-bin/carddisp.pl?gene=HSPA7. Please confirm its role in brain tumors. Based on this article https://pubmed.ncbi.nlm.nih.gov/28012700/, HSP70 family includes HSP72, HSPA6, HSC70, mortalin, GRP78. Please double check and make sure the accuracy.
16. Line 262, 263, “in vivo” should be italic. Same in line 501.
17. Lines 264-267, you are talking about GRP78, then why mention Hsp72?
18. Same lines as above, please double check the reference #70 (https://pubmed.ncbi.nlm.nih.gov/9018110/). The abstract says “The level of hsp72 increased to eight-fold 10 h after hyperthermia alone (44 degrees C for 20 min, D50) and to three-fold 10 h after cisplatin treatment (5 microg/ml) at 37 degrees C for 15 min (D50). In contrast, when the cells were simultaneously heated with cisplatin, the accumulation of hsp72 was suppressed”. I think cisplatin inhibited Hsp72 accumulation. In generally, chemo should inhibit HSP expression. Some studies showed GRP78 mediates cisplatin resistance. Please double check and describe it accurately based on your purpose of statement here.
19. Lines 268-269, GRP78 is mainly presented in ER. Please confirm the presence on the surface of glioblastoma cells. Does this this mean it is located on the plasma membrane?
20. Line 274, do not need to capitalize “Ubiquitin”.
21. Line 292, to be consistent, HSP70 should be Hsp70. Same as HSP90 in line 304, 332.
22. Line 314, is HSP90AA1 correct?
23. Line 351, should use HSPs.
24. The discussion of potential HSP90 inhibitors in section 3 is good. However, based on the writing, most of the drugs showed effectiveness in pre-clinical models (cells or animals). Have any of the inhibitors been approved or in clinical trials?
25. Lines 393-395, HSP990 and BKM120 and radiotherapy showed good clinical outcomes in patients. This is the only drug being described clinical use. More information is needed such as in which clinical phase or approved? If not approved, why?
26. Lines 411-417, 17-DMAG and 17-AAG are geldanamycin (GA) analog. GA was reported as the first HSP90 inhibitor to be evaluated as an antitumor agent. Thus, it will be good to introduce 17-DMAG and 17-AAG together. For more information about HSP90 inhibitors as anticancer drugs, please refer to https://pubmed.ncbi.nlm.nih.gov/34353437/.
27. Line 460, what are plasma membrane-bound Hsp70? Most of Hsp70 are located inside the cells.
28. Line 464, SBD and NBD need full names. If mentioned substrate-binding and nucleotide-binding domains, may need more explanation about structure and function. Otherwise, just mention binding to Hsp70 at different domains.
28. Line 483, “are” should be “is”.
29. Line 503, no need to mention full name of TMZ since it was mentioned previously.
30. In general, HSPs are overexpressed in most of cancer types and have a prognostic role. Brain cancers are the focus of the paper, but do HSPs have a particular role in brain cancers than other types of cancers?
Author Response
We thank the reviewer for the provided comments. We have revised the manuscript accordingly.
COMMENT 1: Line 54, the term “heat shock proteins” is first mentioned here, so the abbreviation of HSP should be here, not in line 56.
ANSWER 1: This was corrected.
COMMENT 2: Lines 61-63, the classification of HSP needs to be more clearly described with additional information, such as adding family and gene names. Adding a summary table of HSP family should be helpful. Please refer to this review article: https://pubmed.ncbi.nlm.nih.gov/28012700/.
ANSWER 2: We have added the Table 1 entitled “Roles of major HSP family members in cell physiology” and referenced the provided manuscript in the text.
COMMENT 3: Lines 64-66. The sentence is confusing. Please rewrite.
ANSWER 3: We have rewritten the sentence.
COMMENT 4: Since the expression of HSPs is mentioned in different types of brain tumors (such as glioblastomas, astrocytoma, etc.), it will be good to provide an introduction of brain tumors such as the classification, grade, prognosis, treatment, etc., especially if there is an association with HSPs. A summary table about the relationship between HSP and brain tumors will be helpful. Or the table can be combined with Table 4.
ANSWER 4: We have added a reference to the latest WHO classification of the CNS tumors in the Introduction section.
COMMENT 5: “et al.” should be italic.
ANSWER 5: We have corrected throughout the manuscript.
COMMENT 6: Line 103, p-HSP27 means phosphorylated HSP27, right? The abbreviation of it is mentioned later line 106. It should be introduced at the first place. Why study of the p-HSP27 is important (needs rationale if it is mentioned)? Any particular role of p-HSP27 in brain tumors compared to regular HSP27.
ANSWER 6: We have corrected this. Additionally, we added: As was shown previously, under different conditions including stress Hsp27 could be phosphorylated that leads to its heterogenous oligomerization with subsequent loss of chaperone activity [39,40]. Presumably, this post-translational modification of p-Hsp27 could influence the tumor cell sensitivity towards applied therapies.
COMMENT 7: Some abbreviations need to give their full name to help readers to understand, such as ATRX, IDH (line 107), PARP (line 149), MCJ (line 195), CCT (line 209), MGMT (line 237) etc. Providing a list of abbreviations will be helpful.
ANSWER 7: We have added the full names of the mentioned proteins.
COMMENT 8: Lines 121-123, figure legend is hard to be understood. Do they miss the unit of 23.28?
ANSWER 8: We have rewritten the figure legend.
COMMENT 9: Immunochemical analysis of HSPs is mentioned in multiple places in the manuscript. Where are HSPs expressed in the brain or brain tumors? Providing an image along with the survival curve should be helpful.
ANSWER 9: We have added the IHC figure (new Figure 6) in the section for HSP70 proteins that demonstrates expression of the chaperone in various stages of tumor progression.
COMMENT 10: Line 162, needs empty space before “In conclusion”.
ANSWER 10: This was corrected.
COMMENT 11: Line 200, it says Hsp60 is expressed higher in tumors than normal tissues. Then it says no difference between other disease tissues (lines 202-204). Does this mean HSP60 is not overexpressed in these diseases? What is the purpose of stating this?
ANSWER 11: This we have rewritten.
COMMENT 12: Line 209, need the introduction of the TCP, CCT members. Are they family members of Hsp60?
ANSWER 12: We have clarified this in the section HSP60.
COMMENT 13: Line 239, what is the Hsp73? The subfamily of Hsp70, HSC70 is ~73kDa? Or misspell of Hsp70?
ANSWER 13: The Hsp70 family of heat shock protiens contains multiple homologs ranging in size from 66-78 kDa, and are the eukaryotic equivalents of the bacterial DnaK. The most studied Hsp70 members include the cytosolic stress-induced Hsp70 (Hsp72), the constitutive cytosolic Hsc70 (Hsp73), and the ER-localized BiP (Grp78). We have added the notion of it into the text.
COMMENT 14: Line 240, gene name HSPA6 should be italic.
ANSWER 14: This was corrected.
COMMENT 15: Line 250. HSPA7 is a pseudogene. Please see https://pubmed.ncbi.nlm.nih.gov/11072087/, https://www.genecards.org/cgi-bin/carddisp.pl?gene=HSPA7. Please confirm its role in brain tumors. Based on this article https://pubmed.ncbi.nlm.nih.gov/28012700/, HSP70 family includes HSP72, HSPA6, HSC70, mortalin, GRP78. Please double check and make sure the accuracy.
ANSWER 15: Indeed HSPA7 is a pseudogene. It was shown that it enables several functions, including ATP binding activity; misfolded protein binding activity; and ubiquitin protein ligase binding activity. Predicted to be involved in several processes, including cellular response to unfolded protein; chaperone cofactor-dependent protein refolding; and protein refolding. https://www.ncbi.nlm.nih.gov/gene?Db=gene&Cmd=DetailsSearch&Term=3311
COMMENT 16: Line 262, 263, “in vivo” should be italic. Same in line 501.
ANSWER 16: This was corrected.
COMMENT 17: Lines 264-267, you are talking about GRP78, then why mention Hsp72?
ANSWER 17: This was corrected.
COMMENT 18: Same lines as above, please double check the reference #70 (https://pubmed.ncbi.nlm.nih.gov/9018110/). The abstract says “The level of hsp72 increased to eight-fold 10 h after hyperthermia alone (44 degrees C for 20 min, D50) and to three-fold 10 h after cisplatin treatment (5 microg/ml) at 37 degrees C for 15 min (D50). In contrast, when the cells were simultaneously heated with cisplatin, the accumulation of hsp72 was suppressed”. I think cisplatin inhibited Hsp72 accumulation. In generally, chemo should inhibit HSP expression. Some studies showed GRP78 mediates cisplatin resistance. Please double check and describe it accurately based on your purpose of statement here.
ANSWER 18: We have checked this.
COMMENT 19: Lines 268-269, GRP78 is mainly presented in ER. Please confirm the presence on the surface of glioblastoma cells. Does this this mean it is located on the plasma membrane?
ANSWER 19: Indeed, several HSP members were shown to be expressed on the surface of the plasma membrane cells, including malignant brain tumors. Reviewed in Shevtsov M, Balogi Z, Khachatryan W, Gao H, Vígh L, Multhoff G. Membrane-Associated Heat Shock Proteins in Oncology: From Basic Research to New Theranostic Targets. Cells. 2020 May 20;9(5):1263. doi: 10.3390/cells9051263. PMID: 32443761; PMCID: PMC7290778.
COMMENT 20: Line 274, do not need to capitalize “Ubiquitin”.
ANSWER 20: This was corrected.
COMMENT 21: Line 292, to be consistent, HSP70 should be Hsp70. Same as HSP90 in line 304, 332.
ANSWER: These were corrected.
COMMENT 22: Line 314, is HSP90AA1 correct?
ANSWER 22: This was corrected.
COMMENT 23: Line 351, should use HSPs.
ANSWER 23: In the beginning of the sentence we decided to use unabbreviated name of HSPs.
COMMENT 24: The discussion of potential HSP90 inhibitors in section 3 is good. However, based on the writing, most of the drugs showed effectiveness in pre-clinical models (cells or animals). Have any of the inhibitors been approved or in clinical trials?
ANSWER 24: To our knowledge no inhibitors have been approved by FDA for clinical application.
COMMENT 25: Lines 393-395, HSP990 and BKM120 and radiotherapy showed good clinical outcomes in patients. This is the only drug being described clinical use. More information is needed such as in which clinical phase or approved? If not approved, why?
ANSWER 25: These agents have not been yet approved for clinical application as further clinical trials are needed.
COMMENT 26: Lines 411-417, 17-DMAG and 17-AAG are geldanamycin (GA) analog. GA was reported as the first HSP90 inhibitor to be evaluated as an antitumor agent. Thus, it will be good to introduce 17-DMAG and 17-AAG together. For more information about HSP90 inhibitors as anticancer drugs, please refer to https://pubmed.ncbi.nlm.nih.gov/34353437/.
ANSWER 26: We have corrected this.
COMMENT 27: Line 460, what are plasma membrane-bound Hsp70? Most of Hsp70 are located inside the cells.
ANSWER 27: Indeed, HSP70 members were shown to be expressed on the surface of the plasma membrane cells, including malignant brain tumors. Reviewed in Shevtsov M, Balogi Z, Khachatryan W, Gao H, Vígh L, Multhoff G. Membrane-Associated Heat Shock Proteins in Oncology: From Basic Research to New Theranostic Targets. Cells. 2020 May 20;9(5):1263. doi: 10.3390/cells9051263. PMID: 32443761; PMCID: PMC7290778.
COMMENT 28: Line 464, SBD and NBD need full names. If mentioned substrate-binding and nucleotide-binding domains, may need more explanation about structure and function. Otherwise, just mention binding to Hsp70 at different domains.
ANSWER 28: These were corrected.
COMMENT 29: Line 503, no need to mention full name of TMZ since it was mentioned previously.
ANSWER 29: This was corrected.
COMMENT 30: In general, HSPs are overexpressed in most of cancer types and have a prognostic role. Brain cancers are the focus of the paper, but do HSPs have a particular role in brain cancers than other types of cancers?
ANSWER 30: Presumably, HSPs in brain tumors function as in other malignant tumors of other localizations. Still there is a lack of data regarding the tumor-specific activity of HSP family members.
Reviewer 2 Report
This is well researched and written manuscript, would provides enough understating for the readers of Cancers journal.
Minor comments:
Line 305-306 paragraph needs to be rephrased
There are few spellings need to be correct.
Author Response
We thank the reviewer for the provided comments. We have revised the manuscript accordingly.
Minor comments:
COMMENT 1: Line 305-306 paragraph needs to be rephrased.
ANSWER 1: We have rephrased the sentence.
COMMENT 2: There are few spellings need to be correct.
ASNWER 2: We have corrected the spellings throughout the manuscript.
Reviewer 3 Report
In this review, the authors presented an overview of the prognostic/diagnostic values of the expression of Heat Shock Proteins in malignant brain tumors and the application of HSP inhibitors for specific targeting of brain tumors. The article is interesting and provides valuable information.
However, the main criticism is that the review is submitted to a Special issue which focuses on ‘novel therapeutic strategies of inducing anti-cancer immune responses or blocking tumor-promoting functions of brain-resident or recruited immune cell types in CNS tumors' and the review in its current form, appears to fall out of the scope of the Special Issue, as there is hardly any material or data on recent developments in HSPs as potential targets for immunotherapies or even as biomarkers for immune surveillance such as that reported previously (please see below).
- Shevtsov M, Multhoff G. Heat Shock Protein-Peptide and HSP-Based Immunotherapies for the Treatment of Cancer. Front Immunol. 2016 Apr 29;7:171
- Taha EA, Ono K, Eguchi T. Roles of Extracellular HSPs as Biomarkers in Immune Surveillance and Immune Evasion. Int J Mol Sci. 2019 Sep 17;20(18):4588.
- Kelly M, McNeel D, Fisch P, Malkovsky M. mmunological considerations underlying heat shock protein-mediated cancer vaccine strategies. Immunol Lett. 2018 Jan;193:1-10.
- Strbo N, Garcia-Soto A, Schreiber TH, Podack ER. Secreted heat shock protein gp96-Ig: next-generation vaccines for cancer and infectious diseases. Immunol Res. 2013 Dec;57(1-3):311-25.
- Baldin AV, Zamyatnin AA Jr, Bazhin AV, Xu WH, Savvateeva LV. Advances in the Development of Anticancer HSP-based Vaccines. Curr Med Chem. 2019;26(3):427-445.
Moreover, there have also been clinical trials reported on HSPs. For instance, when searched under key words “brain cancer” and “heat shock proteins” in the Clinical trials database (Home - ClinicalTrials.gov,), there are 4 entries, examples of which are given below.
- GP96 Heat Shock Protein-Peptide Complex Vaccine in Treating Patients With Recurrent or Progressive Glioma
- HSPPC-96 Vaccine With Temozolomide in Patients With Newly Diagnosed GBM
Other major suggestions for the authors:
- The review lacks mechanistic insights as to how HSPs affect fundamental processes such as apoptosis and autophagy.
- HSPs may also have antiangiogenic effects.
Qi S, Deng S, Lian Z, Yu K. Novel Drugs with High Efficacy against Tumor Angiogenesis. Int J Mol Sci. 2022 Jun 22;23(13):6934.
- HSPs may be involved in chemoresistance.
Lee HJ, Min HY, Yong YS, Ann J, Nguyen CT, La MT, Hyun SY, Le HT, Kim H, Kwon H, Nam G, Park HJ, Lee J, Lee HY. A novel C-terminal heat shock protein 90 inhibitor that overcomes STAT3-Wnt-β-catenin signaling-mediated drug resistance and adverse effects. .Theranostics. 2022 Jan 1;12(1):105-125
Author Response
We would like to thank the reviewer for the provided comments. We have revised the manuscript accordingly.
COMMENT 1: In this review, the authors presented an overview of the prognostic/diagnostic values of the expression of Heat Shock Proteins in malignant brain tumors and the application of HSP inhibitors for specific targeting of brain tumors. The article is interesting and provides valuable information.
However, the main criticism is that the review is submitted to a Special issue which focuses on ‘novel therapeutic strategies of inducing anti-cancer immune responses or blocking tumor-promoting functions of brain-resident or recruited immune cell types in CNS tumors' and the review in its current form, appears to fall out of the scope of the Special Issue, as there is hardly any material or data on recent developments in HSPs as potential targets for immunotherapies or even as biomarkers for immune surveillance such as that reported previously (please see below).
- Shevtsov M, Multhoff G. Heat Shock Protein-Peptide and HSP-Based Immunotherapies for the Treatment of Cancer. Front Immunol. 2016 Apr 29;7:171
- Taha EA, Ono K, Eguchi T. Roles of Extracellular HSPs as Biomarkers in Immune Surveillance and Immune Evasion. Int J Mol Sci. 2019 Sep 17;20(18):4588.
- Kelly M, McNeel D, Fisch P, Malkovsky M. mmunological considerations underlying heat shock protein-mediated cancer vaccine strategies. Immunol Lett. 2018 Jan;193:1-10.
- Strbo N, Garcia-Soto A, Schreiber TH, Podack ER. Secreted heat shock protein gp96-Ig: next-generation vaccines for cancer and infectious diseases. Immunol Res. 2013 Dec;57(1-3):311-25.
- Baldin AV, Zamyatnin AA Jr, Bazhin AV, Xu WH, Savvateeva LV. Advances in the Development of Anticancer HSP-based Vaccines. Curr Med Chem. 2019;26(3):427-445.
Moreover, there have also been clinical trials reported on HSPs. For instance, when searched under key words “brain cancer” and “heat shock proteins” in the Clinical trials database (Home - ClinicalTrials.gov,), there are 4 entries, examples of which are given below.
- GP96 Heat Shock Protein-Peptide Complex Vaccine in Treating Patients With Recurrent or Progressive Glioma
- HSPPC-96 Vaccine With Temozolomide in Patients With Newly Diagnosed GBM
ANSWER 1: We have added the paragraph into the Introduction section concerning the immunological properties of the HSPs with the suggested references.
COMMENT 2: Other major suggestions for the authors:
- The review lacks mechanistic insights as to how HSPs affect fundamental processes such as apoptosis and autophagy.
- HSPs may also have antiangiogenic effects.
Qi S, Deng S, Lian Z, Yu K. Novel Drugs with High Efficacy against Tumor Angiogenesis. Int J Mol Sci. 2022 Jun 22;23(13):6934.
- HSPs may be involved in chemoresistance.
Lee HJ, Min HY, Yong YS, Ann J, Nguyen CT, La MT, Hyun SY, Le HT, Kim H, Kwon H, Nam G, Park HJ, Lee J, Lee HY. A novel C-terminal heat shock protein 90 inhibitor that overcomes STAT3-Wnt-β-catenin signaling-mediated drug resistance and adverse effects. .Theranostics. 2022 Jan 1;12(1):105-125
ANSWER 2: We have added the suggested HSPs effects (i.e., antiangiogenic, involvement into apoptosis and autophagy) into the introduction section with provided references. We have added the reference of Lee et al. Theranostics 2022.
Round 2
Reviewer 1 Report
The authors addressed the reviewers’ comments approximately. The quality of the revised manuscript improves. I recommend it to be accepted for publication.
Reviewer 3 Report
The authors have responded satisfactorily to the comments previously made. I am happy to recommend acceptance of the paper but still maintain that the paper should be under a regular issue but not the special issue where the emphasis is on tumor targeted environment and immunotherapies. Of course, this is only a recommendation and I leave the final decision to the discretion of the Editor.